# Prophylactic Hepatitis E Vaccines: Antigenic Analysis and Serological Evaluation

**DOI:** 10.3390/v12010109

**Published:** 2020-01-16

**Authors:** Yike Li, Xiaofen Huang, Zhigang Zhang, Shaowei Li, Jun Zhang, Ningshao Xia, Qinjian Zhao

**Affiliations:** 1State Key Laboratory of Molecular Vaccinology and Molecular Diagnostics, National Institute of Diagnostics and Vaccine Development in Infectious Diseases, School of Public Health, Xiamen University, Xiamen 361102, Fujian, China; yike_li@163.com (Y.L.); xiaofen_huang@stu.xmu.edu.cn (X.H.); zhigang_zhang@stu.xmu.edu.cn (Z.Z.); shaowei@xmu.edu.cn (S.L.); zhangj@xmu.edu.cn (J.Z.); nsxia@xmu.edu.cn (N.X.); 2State Key Laboratory of Molecular Vaccinology and Molecular Diagnostics, National Institute of Diagnostics and Vaccine Development in Infectious Diseases, School of Life Sciences, Xiamen University, Xiamen 361102, Fujian, China

**Keywords:** antigenic analysis, epitope characterization, hepatitis E vaccine, serological evaluation, virion-like epitopes, well-characterized vaccines

## Abstract

Hepatitis E virus (HEV) infection causes sporadic outbreaks of acute hepatitis worldwide. HEV was previously considered to be restricted to resource-limited countries with poor sanitary conditions, but increasing evidence implies that HEV is also a public health problem in developed countries and regions. Fortunately, several vaccine candidates based on virus-like particles (VLPs) have progressed into the clinical development stage, and one of them has been approved in China. This review provides an overview of the current HEV vaccine pipeline and future development with the emphasis on defining the critical quality attributes for the well-characterized vaccines. The presence of clinically relevant epitopes on the VLP surface is critical for eliciting functional antibodies against HEV infection, which is the key to the mechanism of action of the prophylactic vaccines against viral infections. Therefore, the epitope-specific immunochemical assays based on monoclonal antibodies (mAbs) for HEV vaccine antigen are critical methods in the toolbox for epitope characterization and for in vitro potency assessment. Moreover, serological evaluation methods after immunization are also discussed as biomarkers for clinical performance. The vaccine efficacy surrogate assays are critical in the preclinical and clinical stages of VLP-based vaccine development.

## 1. Introduction

Hepatitis E virus (HEV) belongs to the genus *Orthohepevirus* within the family *Hepeviridae*. HEV was first identified in the 1970s, and its genome was successfully sequenced a decade later [1]. One third of the population in the world may be infected with HEV during their lifetime [2]. HEV infection usually causes acute, self-limiting hepatitis and may cause chronic hepatitis among immunocompromised individuals and solid organ transplant recipients. Moreover, HEV infection poses a threat to pregnant women, with a mortality of 10–50% [3]. A study on the global burden of the disease estimated that approximately 20.1 million people were infected with HEV, leading annually to 3.4 million symptomatic cases, 70,000 deaths and 3000 stillbirths [4]. The burden of HEV-related disease is considerable in developing countries, especially in Africa and Asia [5]. Although there is no evidence of outbreaks of hepatitis E in developed countries, sporadic locally acquired cases of HEV infections have been reported in France [6], Germany [7], Switzerland [8], Australia [9], Japan [10] and so on. To date, eight genotypes of HEV have been isolated, and at least five genotypes (genotypes 1–4 and 7) can cause human infection [11,12,13].

HEV contains a single-stranded, positive-sense RNA with a size of approximately 7.2 kb. The genome of HEV comprises three open reading frames (ORFs) [14]. ORF1 encodes a non-structural protein that is responsible for viral RNA replication [15]. Recently, a protein encoded by the newly discovered ORF4 (within ORF1) was shown to be able to stimulate the replication of genotype-1 HEV [16]. ORF3 encodes a phosphoprotein, which is involved in virus release from host cells [14]. HEV was recently found to be a quasi-enveloped virus. It exists as non-enveloped virions in faeces and urine for transmission, whereas its form could be predominantly enveloped in serum for evading neutralizing antibodies [17,18,19]. The existence of two different forms of virions might be related to the function of ORF3 [20]. Notably, the most well-studied ORF, ORF2, encodes the sole capsid protein (pORF2), which has the ability to self-assemble into viral capsids to package the viral RNA after each replication cycle. Therefore, the capsid protein pORF2 is a rational target for vaccine design [21].

Based on pORF2, different truncations using recombinant DNA technology were performed to yield various proteins with different assembly forms [22]. Some of the truncated proteins form into virus-like particles (VLPs) or subviral particles, presenting virion-like epitopes on the particle surface [23]. In this review, we use the term VLP in a loose term regardless of the size of the VLPs. Some VLPs could be more virion-like, including the size and the array of the epitope, while others could be smaller in size and less regular in particle formation, which are essential for effectively stimulating the immune response. Among these truncated pORF2 molecules, three vaccine candidates have been studied in clinical trials. One vaccine, with a trade name of Hecolin^®^, was licensed in China in 2011. However, it was not prequalified by the World Health Organization (WHO), which was necessary for introduction into countries where the disease burden was considerable [24]. The WHO issued a recommendation in 2018 to provide guidance to national regulatory authorities and manufacturers on the manufacturing process and on nonclinical and clinical aspects to assure the quality, safety and efficacy of recombinant hepatitis E vaccines [3]. Needless to say, quality assurance is of utmost importance for a licensed vaccine for widespread use. For the VLP-based vaccines, the presence of virion-like epitopes on the surface of VLPs is the structural basis to elicit protection against the specific pathogen [25]. The process comparability, analytical comparability, and the link of a change to the clinical outcome were considered as essential issues for a VLP-based vaccine. One A-vax case study discussed the risk assessment and control strategy, which supports comparability studies of vaccines [26]. Various methods have been established as a “toolbox” to evaluate the vaccine antigen for quality assurance during bioprocessing and for stability during storage and transportation. Among them, a large panel of monoclonal antibodies (mAbs) against the HEV capsid protein was developed, and assays based on these mAbs are used during bioprocessing and in lot release and stability testing [25]. In addition, immunization with a vaccine could induce a response against viral B cell epitopes in vaccine recipients [27]. The functional polyclonal antibodies in serum elicited by immunization can be analysed by a series of methods. In this review, we focus on epitope-specific antigenic analyses of hepatitis E vaccine antigen and evaluation assays of vaccine-elicited functional antibodies for its overall neutralization activity or against well-characterized epitopes.

## 2. The Design of Hepatitis E Vaccines

### 2.1. Molecular Structure of Different Truncated Versions of pORF2

HEV ORF2 encodes a viral capsid protein containing 660 amino acids (aa). A recent study reported that a secreted form of pORF2 was observed in serum samples from both HEV-infected rhesus macaques and humans. Two different forms, pORF2^C^ (capsid) and pORF2^S^ (secreted), are two different translation products of the same viral ORF2 gene [28]. Compared to ORF2^C^, ORF2^S^ contains an additional 15 aa. This 15-aa peptide segment could represent a signal sequence that drives ORF2^S^ secretion [28]. The prolonged existence of ORF2^S^ in the blood raises the possibility of decoying, leading to partial or full depletion of neutralizing antibodies in the sera of convalescent patients. Thus, ORF2^S^ antigen in HEV-infected patient serum could reduce the protective efficiency after vaccination. Further characterization of ORF2^S^, such as the complexing forms and kinetics in blood is of clinical importance [28].

To test the immunogenicity of the capsid protein, a series of truncated forms of pORF2 were prepared in different laboratories [22]. They include the nearly full-length p595 (aa 14−608) and p495 (aa 112–606), as well as much smaller particulate forms, such as p239 (aa 368–606) and p179 (aa 439−617), and the even shorter proteins, such as E2 (aa 394–606) and E2s (aa 459–606), that are amenable for crystallization (Figure 1A). The N- and C-terminally truncated version of pORF2, i.e., aa 14−608 (p595), can form T = 3 icosahedral VLPs that are highly analogous to native virions [29]. Additional highly virion-like VLPs were observed with p495, with a further truncation from p595 containing aa 112−606. P495 self-assembled into well-formed T = 1 icosahedral VLPs [30]. Through analysis of pORF2 using cryo-EM and X-ray crystallographic studies, three functional domains (S, P1 and P2) were identified [29]. The P2 domain, also named the E2s domain, harbours the major neutralizing epitopes on the viral capsid. The E2s domain, which has available high-resolution structural data, forms a tight homodimer with key interacting residues identified at the dimeric interface [31]. The E2s domain may play a key role in the interaction of HEV with host cells because antibodies directed towards this region tend to be neutralizing and functional. Moreover, the structure of the E2s domain in a complex with a potent neutralizing mAb, 8C11 [32], was determined. The epitopes recognized by the mAb 8C11 were identified to comprise three different peptide segments, namely, Asp496-Thr499, Val510-Leu514, and Asn573-Arg578. Among these, Arg512 was found to be the most crucial residue for 8C11 interaction with and neutralization of HEV. In addition, the complex crystal structure of 8G12 [33], another highly efficient neutralizing mAb, with E2s was also determined. Several important residues (Glu549, Lys554 and Gly591) were revealed that played an important role in 8G12 neutralization.

A 66-aa extension from the N terminus of E2s and could stabilize E2 (aa 394−606) to form hexamers in solution [34]. Further extended with another 26-aa extension on E2, p239 (aa 368–606) could self-assemble into a particulate form or VLPs with a 20–30 nm diameter [35]. Although there is a certain degree of particle size and particle regularity, the immunogenicity of the p239-based antigen was quite impressive. Notably, in experiments conducted in parallel, the immunogenicity of p239-based VLPs was shown to be approximately 240 times stronger than that of E2 in mice [35]. The enhanced immunogenicity is likely due to the multiplicity of arrayed virion-like epitopes on the VLP surface. More recently, p179 (aa 439–617) was found to self-assemble into VLPs with a diameter of approximately 20 nm [36]. This VLP form of the antigen also shares the major neutralization epitopes with p239. This antigen, along with the previously mentioned p239 and p495, was tested in clinical trials with the goal of human vaccine development (Figure 1B).

### 2.2. Hepatitis E Vaccines

While more than fourteen HEV vaccine candidates have been studied [37], only three candidates have progressed to clinical trials (Figure 1B). Aluminium-based adjuvants were used in the formulation of all three vaccine candidates. The p495-based vaccine was the first vaccine candidate evaluated in clinical trials with the support of GlaxoSmithKline. It was produced in insect cells with a baculovirus expression system. Desirable safety and efficacy in a phase II clinical trial study were demonstrated for the p495-based vaccine. The vaccine efficacy was 95.5% after three immunizations. In addition, the reports of any adverse event were similar in vaccine group and placebo group [38]. However, this project did not progress further after the phase II clinical trial possibly due to the lack of commercial value. Another vaccine antigen based on p239 was expressed in an *Escherichia coli* (*E. coli)* expression system. The p239-based vaccine was developed and licensed, with safety and efficacy demonstrated in a large-scale phase III clinical trial. The efficacy was 100% over 12 months in preventing hepatitis E among participants receiving all three doses of the immunization [39]. Moreover, a follow-up study showed that immunization with the p239-based vaccine could provide long-term (up to 4.5 years) protection against hepatitis E, with an efficacy of 86.8% [40]. A post-licensure study showed that the p239-based vaccine was immunogenic and well tolerated in the elderly population (>65 years old), setting the stage for expanded recommendation of the vaccine to the aged populations in whom HEV infection could be more harmful [41]. More recently, a p179 (expressed in *E. coli*)-based vaccine candidate was tested in a phase I clinical trial. This study showed safety and good tolerance for the 16- to 65-year-old population [36] (Figure 1B), and a phase II clinical trial is ongoing. The recreation of the neutralizing epitopes on the truncated pORF2 forms as vaccine antigens is the key for assuring the elicitation of functional antibodies. The existence of these virion-like epitopes can be characterized using a series of immunochemical techniques.

## 3. Analysis of the Native-Like Epitopes on Recombinant Antigens

A recombinant VLP-based antigen is an ideal candidate for use in vaccine formulation due to its high immunogenicity and desirable safety performance. Comprehensive analysis of a vaccine antigen is important for the quality assurance of a vaccine. Therefore, it is essential to develop a series of methods to evaluate the vaccine antigen during processing, formulation and storage/transportation. Various methods, such as biophysical, biochemical, immunochemical and in vivo potency assays, were established and applied to two licensed recombinant VLP-based vaccines: hepatitis B vaccine and human papillomavirus vaccine [42]. For the hepatitis E vaccine, the toolbox was composed of various analytical approaches, especially specific epitope-based immunochemical assays, which will be discussed in the following section. Among all the critical quality attributes of prophylactic vaccines, the presence and the integrity of the virion-like epitopes on the recombinant antigens are critically important for eliciting functional antibodies and conferring protection against viral infection [25].

### 3.1. Epitope-Specific Analysis of the HEV Vaccine Antigen by mAbs

The presence of clinically relevant epitopes on the VLP surface is normally the structural basis for eliciting functional antibodies against viral proteins against the viral capsid. For HEV, the E2s domain, an important part of pORF2, was shown to harbour the major neutralizing epitopes. To study the clinically relevant epitopes, recombinant p239 protein was used as an immunogen for the preparation of a large panel (96) of murine monoclonal antibodies (mAbs) [43]. Among them, approximately 50% of the mAbs were chosen for further analysis due to their significant reactivity with p239. Overall, 20% of the mAbs recognized linear epitopes (such as 3A11, 16D7, 12A10, *etc*.), and 30% of the mAbs recognized conformational epitopes (such as 8G12, 8C11, 9F7, *etc.*) on the E2s domain. There are 23 conformation-dependent mAbs that were selected for further study due to their ability to capture authentic HEV virions in vitro. Six distinct conformation-dependent epitopes (C1-C6) were identified in the E2s domain by clustering analyses [43]. As a representative conformational and neutralizing antibody, 8G12 could react with p239 dimer and capture HEV virions. The epitopes recognized by 8G12 were located around the E2s dimerization interface, and the key epitope residues were Glu549 and Gly59 in the E2s domain [33]. Another neutralizing antibody, 8C11, recognized C5 epitopes, which are located in the groove zone of the E2s domain, away from the dimeric interface [32]. Due to the neutralizing activity of these two well-characterized mAbs, 8C11 and 8G12, their targets likely represent two distinct functional epitopes on the HEV viral capsid. These epitopes are likely the clinically relevant epitopes. Thus, based on 8C11, 8G12 and other mAbs, various immunochemical assays were developed. These assays, along with a battery of other physico-chemical assays, have been used to evaluate the critical quality attributes of the vaccine products.

Surface plasma resonance (SPR)-based BIAcore is a one-site binding assay probing the antibody-antigen interaction on a sensorchip in real time. The assay is label-free, with no need to label the antibody or the antigen. Due to the high throughput and the degree of automation, the antigenicity for multiple batches of p239 was measured using a panel of 5 mAbs. They included mAbs that recognize linear epitopes (12A10 and 3A11) and mAbs that recognize conformational epitopes (8G12, 8C11 and 12F12). Good lot-to-lot consistency and desirable stability of the HEV p239 antigen was demonstrated [25]. Another one-site binding assay is a solution competition enzyme-linked immunosorbent assay (ELISA), where one mAb is used for interrogating the antigen immune reactivity in solution. As an example, the solution interaction of recombinant HBsAg and the mAb 5F11 was monitored for quality analysis of the antigen [44]. This assay could detect subtle differences in antigen epitopes in solution, avoiding potential conformational changes during the surface adsorption process [45]. Using this solution competition ELISA, comparable antigenicity among different lots of p239 was demonstrated with multiple antibodies, namely, 8C11, 8G12, 9F7, 12A10, etc. [25].

Vaccine formulation in general contains particulate-form adjuvants, making antigenicity more difficult. Recently, a new in situ antigenicity analysis method, high content analysis (HCA), which is capable of antigenicity analysis of aluminium-adsorbed antigen without the need for dissolution, has been developed [46]. Using fluorescence-labelled 8G12, the real-time stability of multiple lots of p239-based vaccines was demonstrated after long-term storage. HCA-based in situ analysis is a fluorescence-based measurement. Thus, it enabled the simultaneous detection of two or more differently labelled mAbs. These could be multiple mAbs against the same antigen or against different antigens in the vaccine formulation. Zhang et al. [47] reported the antigenic analysis of VLP-based antigens adsorbed on adjuvants without dissolution using two distinctly fluorescence-labelled mAbs. HEV VLP-based vaccines or HEV antigens in combination vaccines could be analysed using technology with multiple detection antibodies. In addition, based on two-site binding, a highly sensitive and robust sandwich ELISA with the mAb 3A11 as the capture antibody and the mAb 8C11 as the detection antibody was developed for routine antigenicity testing. Comparable antigenicity of p239 pre- or post-dissolution treatment was determined, showing no change in epitopes after proper dissolution treatment with adjuvants [48]. Meanwhile, multiple lots of hepatitis E vaccines retained antigenicity after 36 months of storage [46]. As a robust technical assay, the sandwich ELISA has the potential to become a product release assay since this assay format is amenable for set up in a manufacturing setting.

### 3.2. The Application of an In Vitro Relative Potency Assay (IVRP)

The potency assay is the most critical for vaccine characterization, comparability evaluation and lot-release testing. Currently, potency assays for vaccines can be roughly divided into in vivo animal-based potency assays and in vitro relative potency assays [49]. Traditional evaluation of vaccine potency is an in vivo animal-based potency assay, which is the closest mimic of a human response to a vaccine. It is widely used in the preclinical development stage of a vaccine. However, in vivo animal-based potency assays are time-consuming (4–6 weeks) and exhibit poor precision and high relative standard deviations. In addition, based on the “3R” principle (reduction, replacement and refinement of animals), it is highly desirable to establish an alternative in vitro potency assay to assess the binding activity of the antigen to functional antibodies during bioprocessing [26,50]. Therefore, additional alternative in vitro assays could be implemented to minimize animal use. With good reproducibility and robustness, the sandwich ELISA has the potential to be a candidate in vitro relative potency assay for product release [51]. Notably, Sandwich ELISA was chosen as in vitro alternatives to evaluate the potency of human rabies vaccines and was accepted for an international collaborative study [52]. In addition, it has been reported that IVRP has a good correlation with animal-based potency assays for licensed human papillomavirus and hepatitis B virus vaccines [53,54]. For the hepatitis E vaccine, the correlation between animal-based efficacy and IVRP needs to be evaluated in the future. In general, a mouse potency assay, as indicated by an ED_50_ value, is a potency assessment for a vaccine formulation to cause seroconversion in the sera of the test animals. Either binding antibodies or neutralizing antibodies could be evaluated via such a seroconversion analysis. This fact is also true for clinical serological testing using either binding titres or neutralization titres. While the latter is more desirable, its limited throughput would normally exclude its use in support of clinical sample testing.

## 4. The Serological Evaluation

Like most prophylactic VLP-based vaccines, the hepatitis E vaccine elicits a strong humoral response. Effective presentation with orderly arrayed epitopes on the antigen surface and high local epitope density could account for the effective B cell response. The generation of neutralizing antibodies against clinically relevant epitopes is the underlying mechanism of efficacious prophylactic vaccines [49]. In general, the titre of neutralizing antibodies elicited by vaccination correlates with the efficacy of the specific vaccine. In addition, the p239-based hepatitis E vaccine could also stimulate a cell-mediated immune response. Khateri et al. [55] demonstrated via IFN-γ ELISPOT assay that p239 vaccination was able to induce cellular immunity. However, due to limited research, the contribution of cellular immunity to protection against HEV infection is not clear. In the following section, a series of quantitative methods for evaluation of the B cell response elicited by the hepatitis E vaccine are discussed.

### 4.1. The Binding Antibody Analysis

While the clinical end point for vaccine efficacy in a clinical trial could be a disease or a pathological marker, the seroconversion induced by vaccines is always a quantitative index to be measured, owing to the vaccination with a test vaccine. The binding titre in serum samples in an ELISA is always a straightforward method for serological analysis [56]. In ELISAs for measuring titres, 3 recombinant proteins and VLPs were used as coating antigens. For the hepatitis E vaccine, the coating antigens are generally the same or similar proteins as the vaccine antigens derived from ORF2 proteins. For the p239-based vaccine, anti-HEV IgG in serum samples from vaccinated humans and rhesus monkeys was analysed via E2-coated ELISA [57] (Table 1). For another vaccine candidate based on p495, the total immunoglobulin in serum samples derived from cynomolgus monkeys after immunization was evaluated in a p495-coated ELISA. Anti-HEV antibodies in monkeys’ serum samples after receiving two doses of p495-based vaccine were elicited with titres of 1:10^4^, and monkeys developed neither hepatitis nor viremia when challenged with virus [58].

Since different vaccine developers use different coating antigens, standardization of the serological assay would facilitate cross-lab and cross-product comparison if the same coating antigen could be used in the assay. Wen et al. [59] investigated the immunogenicity differences between hepatitis E vaccines based on p239 and p179. HEV p166 (aa 452−617) was used as a coating antigen to evaluate anti-HEV IgG in the serum of mice and humans immunized with p179 or p239 [59]. However, p166 was composed of aa 452−617, which is much shorter than p239 and may exclude some important functional epitopes. With this caveat in mind, a p495-based VLP antigen was more virion-like and contained all functional epitopes of p179 and p239 (Figure 1A), making it a better coating antigen. Similarly, efforts are being made to standardize the human papillomavirus serological assays using a more native- or virion-like VLPs as coating antigen in the ELISAs. Although the cross-lab comparability of ELISAs for human papillomavirus−16 and −18 has been formed, international standards are still required for the additional types in Gardasil^®^9 [56]. In most cases, correlations were observed between the binding titres and neutralizing titres. This result supports the use of binding titre measurement to support the clinical trials while using a subset of the clinical samples to establish such a correlation.

### 4.2. The Neutralizing Antibody Analysis

An evaluation method of virus neutralizing efficiency of serum samples is essential to assess the vaccine-elicited immune response against the virus. However, inefficient propagation of HEV in cell models has posed a challenge to the evaluation of neutralizing antibodies in sera. The propagation and production of HEV were attempted in primary hepatocytes. Based on this cell system, an in vitro neutralization assay was developed to evaluate the anti-HEV antibodies. However, the difficulty in culturing the cells has limited its application [60]. To date, at least two cell culture systems for HEV have been developed. They are the culture system in PLC/PRF/5 and A549 cells established by Tanaka et al. [61] and the culture system in HepG2/C3A cells established by Shukla et al. [62] (Table 2). These cell culture systems allow the virus to propagate enough authentic HEV virions to support the neutralizing analysis [63]. Based on these HEV cell culture systems, polyclonal antibodies in serum samples or mAbs derived from immunized animals showed neutralizing ability against different viruses (Table 2). In one study, the HEV genotype 3 strain Kernow was shown to have high replication efficiency, using HepG2/C3A as a cell substrate. The infectivity of HEV was inhibited efficiently by the serum samples from immunized individuals [64] (Table 2). Apart from polyclonal antibodies in serum, several mAbs showed neutralizing ability. The neutralizing activity of two represent mAbs (8G12 and H6225) was studied with different cell systems. Gu et al. [33] reported that 8G12 was able to inhibit the replication of HEV based on Huh7 cells. H6225 was able to neutralize a genotype 3 HEV strain (JE03-1760F) in PLC/PRF/5 cells [65] (Table 2). Although many improvements were made, there was still a lack of an efficient and reliable cell-culture system for an HEV virus neutralization assay. Therefore, finding a surrogate of native virions may be a good choice to develop cell-based assays as HEV serological assays for vaccine evaluation.

The vaccine antigen p239 was used as a surrogate for native HEV virions in a cell-based functional assay for antibodies. He et al. [70] used p239 to simulate native HEV for virus attachment onto hepatocytes. P239 could attach to and enter the cells of four susceptible cell lines, i.e., HepG2, Huh7, PLC/PRF5 and A549 [71]. When neutralizing mAbs were used, the cell attachment of p239 was effectively blocked. More recently, a “neutralizing-like” blocking assay based on HepG2 cells was developed by Cai et al. [69] for antibody functionality assessment (Table 2). The assay was based on biotin-conjugated p239 and staining with allophycocyanin-conjugated streptavidin to amplify the fluorescence signal. Using this assay, the p239-blocking activity of serum samples from HEV-infected and vaccinated macaques was quantitatively evaluated [69].

With a well-characterized murine mAb (8G12), a competitive ELISA was developed for functional antibody evaluation of serum samples. 8G12 could efficiently block the binding of polyclonal antibodies in immunized human and rhesus macaque serum to vaccine antigen. 8G12 was applied to develop a competitive ELISA assay to detect 8G12-like antibodies in mouse and human serum samples. The 8G12-like antibody was predominant among the vaccine-induced anti-HEV antibodies in both human and mouse sera. Therefore, 8G12-like antibodies might be a promising surrogate for neutralizing antibodies and have the potential to be used as an indicator of the neutralization ability of the hepatitis E vaccine [72]. In another study, the mAb 8C11 was also used in a competitive ELISA to evaluate 8C11-like antibodies in serum samples from immunized mice [73]. In a similar format but with another virus, Palivizumab-like antibodies against respiratory syncytial virus were tested via a competitive ELISA to reflect the neutralization antibodies after vaccination [74]. Similarly, in the case of human papillomavirus, the neutralizing antibody level of each human papillomavirus serotype elicited by vaccination was assessed by a multiplex competitive Luminex immunoassay with functional and type-specific mAbs (such as H16.V5 for type 16) as a specific probe for each type [75].

Recently, some modifications of HEV were performed at the genetic level to quantitatively monitor the infection and replication of HEV. Swiss scientists claimed that HEV genomes harbouring a haemagglutinin epitope tag or a small luciferase were found to be fully functional. This approach could enable the efficient production of infectious viruses with specific tags for ease of virus quantitation [76]. Based on HepG2/C3A cells infected with the HEV genotype 3 strain, virus replication and infection were monitored efficiently. Since these viruses can be easily engineered, different genotypes of HEV strains should be prepared in parallel, facilitating evaluation of the cross-genotype virus neutralization efficiency of various serum samples.

## 5. Can Current Vaccines Protect against Hetero-Genotypes?

To date, three VLP-based hepatitis E vaccines (p495-, p239-, and p179-based vaccine) have been tested in human volunteers and showed desirable safety and efficacy. Each vaccine antigen was derived from one given virus genotype. However, there are at least five genotypes (genotypes 1–4 and 7) that are capable of infecting humans. In addition, various HEV isolates have been identified from diverse animal species. Recently, rabbit HEV of genotype 3ra was found to be capable of infecting humans in Switzerland [8]. With different circulating strains of HEV in different continents, a question is always asked—is the current vaccine containing one antigen from a given genotype capable of protecting against other different genotypes?

In clinical trials conducted in China, where the majority of HEV isolated from patients is genotype 4, the efficacy of a p239-based vaccine was evaluated by a large-scale randomized double-blind phase III trial. The vaccine antigen derived from genotype 1 could elicit a robust antibody response in most patients, showing excellent cross-genotype protection against HEV [39] (Table 3). While the clinical efficacy data against genotypes 1, 2 and 3 are not available, the cross-genotype protection of the hepatitis E vaccine was also determined in preclinical tests. The p239-based vaccine was efficacious in preventing infection in rhesus monkeys challenged with genotype 1 or genotype 4 virus [35]. For the p495-based vaccine, the cross-genotype protection among non-human primates was also determined among HEV genotypes 1, 2 and 3 [77]. Furthermore, the p179-based vaccine derived from genotype 4 showed complete cross-protection against genotype 1 and genotype 4 virus in rhesus monkeys [36].

In addition, a series of cell-based analyses also showed cross-genotype protection against HEV. The infection of Kernow (genotype 3) was blocked by vaccinated human serum (genotype 1) based on HepG2 cells, suggesting that the genotype 1 HEV vaccine was able to protect against genotype 3 virus infection [64]. In addition, the mAb 8G12 could inhibit genotype 1 and genotype 4 HEV infections in the Huh7 cell model [33] (Table 3). At the biochemistry level, according to the crystal structure of different forms of truncated pORF2, the protrusion domains (E2s domain) from different genotypes show highly similar structural features. The crystal structures of the immune complexes of 8G12 and the E2s domains (genotypes 1 and 4) were very similar. Thus, 8G12 could efficiently bind to the E2s molecules derived from all 4 different types [33]. All lines of accumulating evidence from serological, biochemical and structural studies indicate that the protrusion domain of the viral capsid plays an important role in protection. These functional epitopes in association with these protrusions are immuno-dominant and are highly conserved across different HEV genotypes. It is conceivable that vaccination with an antigen derived from a given genotype could confer protection from infection by all the HEV genotypes.

## 6. Conclusions and Prospects

Prophylactic vaccination is an effective method to protect against HEV infection and control HEV infection-associated diseases. The ordered arrangement of the clinically relevant epitopes is the structural basis of a VLP-based hepatitis E vaccine. A multifaceted and comprehensive toolbox composed of orthogonal methods was established to ensure vaccine quality, safety and efficacy. Similar analytical assays, built with the same concept, could also be used for other VLP-based vaccine antigens. Serological evaluation after vaccination should be standardized to allow cross-laboratory comparison when multiple developers are producing vaccines against the same indication.

Outbreaks of HEV infections in low-resource regions continue to pose challenges to public health. Outbreaks were reported in at least 30 countries in Africa, Asia, and North America in recent years. During the past decade (2009−2019), six outbreaks of hepatitis E, each causing at least 1000 infections, were reported in African countries, including South Sudan, Namibia, Uganda, Nigeria, Niger and Chad. The most recent and ongoing (since 2017) HEV outbreak in Namibia has caused at least 4669 infections and 41 deaths [78]. In addition to the outbreaks in developing regions, the high rate of asymptomatic HEV infections worldwide has raised concern of infection via blood donation in recent years. For instance, HEV IgG antibody seroprevalence in blood donors was found to be 19.8% in Denmark, owing to the better awareness and improved assays [79]. Although most patients remain asymptomatic after accepting infected blood product, those immune-suppressed individuals take a considerable risk of developing chronic HEV infection. Use of the licensed vaccine should be further promoted to enhance coverage, particularly for the outbreak-prone regions or among high-risk populations, such as pregnant women or immuno-suppressed or immune-compromised individuals. Recently, the protective effect of the hepatitis E vaccine for pregnant women was evaluated in Bangladesh (NCT02759991) with support from the Norwegian Institute of Public Health, showing no safety concerns for the participants related to immunization [80]. Furthermore, another safety clinical trial of the hepatitis E vaccine was initiated in May 2019 in a healthy US adult population (NCT03827395) with the support of the National Institute of Allergy and Infectious Disease [81]. With efforts from multiple fronts towards the control of this vaccine-preventable disease, the effect of this public health tool should be maximized in the future.

## Figures and Tables

**Figure 1 viruses-12-00109-f001:**
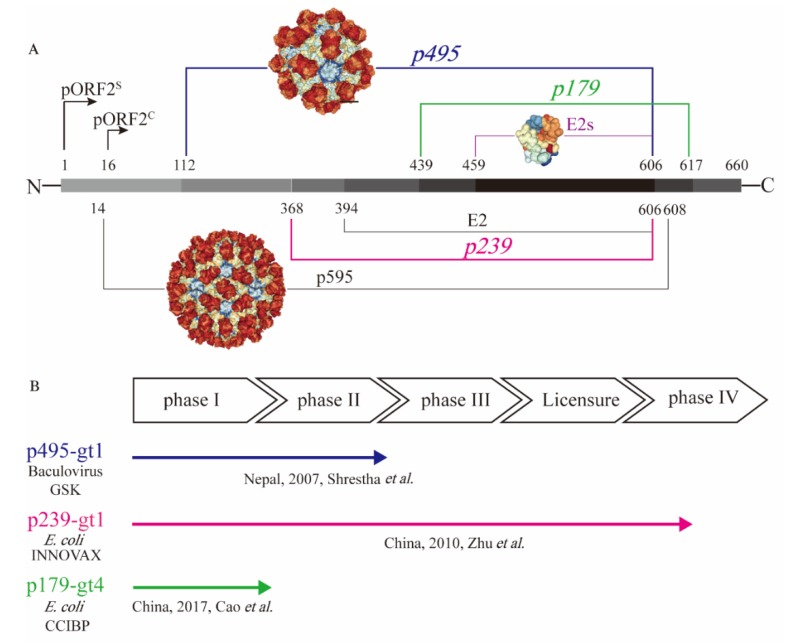
Presentation of different truncated versions of hepatitis E virus (HEV) pORF2. (**A**) shows the molecular structure of truncated pORF2, and (**B**) shows three existing HEV vaccines, which have been studied in clinical trials. HEV pORF2 consists of 660 amino acids. HEV p595 (aa 14–608) can form a virus-like particle (VLP) that is similar to the native virion. The structure of p595 was demonstrated by cryo-EM. HEV p495 (aa 112–608) can form a VLP, and the structure has been determined by X-ray. HEV p495 was used as a vaccine antigen manufactured by GSK, which showed good safety and efficacy in a phase II clinical trial. HEV p239 (aa 368–606), named Hecolin^®^, has been licensed in China. The HEV p179 (aa 439−617)-based vaccine, which was manufactured by Changchun Institute of Biological Products Co., Ltd. (CCIBP), was safe and well tolerated in a phase I clinical trial. E2 was a useful candidate for diagnostic reagents and was able to form hexamers in solution. The structure of E2s (aa 459–606), the shortest version to form a dimer harbouring the major neutralizing epitopes, was determined at a high resolution.

**Table 1 viruses-12-00109-t001:** Different antigens used in the ELISA methods for supporting the vaccine clinical trials or preclinical development.

Vaccine Antigen.	Antibody Type	Species	Coating Antigen	Reference
p495	total immunoglobulin	rhesus monkey	p495 (112−606)	Tsarev et al., 1997 [58]
		human		Shrestha et al., 2007 [38]
p239	IgG	rhesus monkey	E2 (394−606) *	Li et al., 2005 [35]
		human		Zhu et al., 2010 [39]

* E2 and p239 share most of the critical epitopes.

**Table 2 viruses-12-00109-t002:** The neutralization analysis of monoclonal or polyclonal antibodies after vaccination or infection.

Host Cell	Virus/Genotype	Type of Sample	Reference
***Native virus***			
HepG2/C3A cells	Kernow (gt3)	serum (rhesus macaque & human)	Liu et al., 2019 [64] *
Huh7 cells	stool and bile-derived HEV (gt1/4)	mAb 8G12	Gu et al., 2015 [33]
PLC/PRF/5 cells	JE03-1760F (gt3)	mAb H6225	Takahashi et al., 2008 [65]
Primary hepatocytes	Burma (gt1)	purified IgGs (cynomolgus monkey)	Tam et al., 1997 [66]
PLC/PRF/5 cells	F23, SAR-55 (gt1)	serum (cynomolgus monkey & human)	Meng et al., 1997 [67]
HepG2/C3A cells	Sar55, Mex14, Meng (gt1)	serum (rhesus macaque)	Emerson et al., 2006 [68]
PLC/PRF/5 and A549 cells	JE03-1760F (gt3)	serum (human)	Tanaka et al., 2007 [61]
PLC/PRF/5 and A549 cells	serum-derived HEV (gt3)	serum (human)	Takahashi et al., 2010 [20]
***Virus surrogate***			
HepG2 cells	recombinant protein 239	serum (rhesus macaque)	Cai et al., 2016 [69]

“*” Using similar cell lines and virus strain, a robust HEV infection and replication system was reported recently (Todt et al. Robust hepatitis E virus infection and transcriptional response in human hepatocytes. *Proc Natl Acad Sci* 2020) with high virus titres obtained consistently. It is conceivable that the use of this system should facilitate the development of a virus neutralization assay.

**Table 3 viruses-12-00109-t003:** The protection of the hepatitis E vaccine among different virus genotypes.

Vaccination	Subject	Genotype 1	Genotype 2	Genotype 3	Genotype 4	Reference
***Animal- or human-based analysis***						
p495-gt1	human	√	-	-	-	Shrestha et al., 2007 [38]
p239-gt1	human	-	-	-	√	Zhu et al., 2010 [39]
p495-gt1	rhesus monkey	√ a	√ b	√ c	-	Purcell et al., 2003 [77]
p495-gt1	rhesus monkey	√ a	√ b	-	-	Tsarev et al., 1997 [58]
p239-gt1	rhesus monkey	√	-	-	√	Li et al., 2005 [35]
p179-gt4	rhesus monkey	√	-	-	√	Cao et al., 2017 [36]
***Cell-based analysis***						
p239	HepG2/C3A	-	-	√ d	-	Liu et al., 2019 [64]
8G12	Huh7 cells	√	-	-	√	Gu et al., 2015 [33]

“-” means no data; “√” means the cross-genotype protection; “a” SAR-55 (GenBank accession no. AF444002); “b” Mex-14 (GenBank accession no. M74506); “c” US-2 (GenBank accession no. AF060669.1); “d” Kernow (GenBank accession no. JQ679013).

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
