# Peer review of "Prophylactic Hepatitis E Vaccines: Antigenic Analysis and Serological Evaluation"

_viruses, 2020, doi:10.3390/v12010109_

Round 1
Reviewer 1 Report
The manuscript (Viruses-677621) provides an overview of the current HEV vaccine pipeline and future development with the emphasis on defining the critical quality attributes for the well-characterized vaccines. This review is well written and updated.
Minor Comments:
Page 8, line 303: please, delete recently (2016 is not recently).
Page 8, line 326: the reference 55 should not be cited here
Table 2: Put in capital letters: “Native virus” and “Virus surrogate”
Please, explain in the text the meaning of “Native virus” and “Virus surrogate”
Page 9, line 351: this reference is not in the list of references: He et al.
Please, review all references. Authors should abbreviate the journal name according to instructions.
Reference 66: lowercase letter reference tittle.
Author Response
We thank the reviewer for his/her careful review of the manuscript. We do appreciate the nice words on the manuscript and the constructive comments on improving the writing. We have made changes to the manuscript based on these comments (with track-change mode), which have resulted in a much improved manuscript. Line to line responses are listed below.
Comments and Suggestions for Authors
The manuscript (Viruses-677621) provides an overview of the current HEV vaccine pipeline and future development with the emphasis on defining the critical quality attributes for the well-characterized vaccines. This review is well written and updated.
Point: Page 8, line 303: please, delete recently (2016 is not recently).
Response: Thanks for pointing this out. We have deleted the word “recently” in the revised manuscript per suggestion.
Point: Page 8, line 326: the reference 55 should not be cited here
Response: Thanks for the careful review by the reviewer. This reference is now cited in Line 349 in the revised manuscript. “These cell culture systems allow the virus to propagate enough authentic HEV virions to support the neutralizing analysis [63]”.
Point: Table 2: Put in capital letters: “Native virus” and “Virus surrogate” Please, explain in the text the meaning of “Native virus” and “Virus surrogate”
Response: Thanks for the comments. We have made the revision accordingly. Native virus means that the virus were obtained from animal feces or gained by cultivated in vitro. In this reference, p239, which was used as a virus surrogate, is compose of protein rather than an authentic virus.
Point: Page 9, line 351: this reference is not in the list of references: He et al. Please, review all references. Authors should abbreviate the journal name according to instructions.
Response: We thank the reviewer for his/her careful review. We have inserted this reference in Line 376 of the revised manuscript as Reference 70. In addition, we have carefully checked all of abbreviations of the journal names in revised manuscript.
Point: Reference 66: lowercase letter reference tittle.
Response: We thank the reviewer for his/her careful review and for pointing this out. We have made the changes in the revised manuscript as Reference 75.
Reviewer 2 Report
Li et al present a review about current knowledge and perspectives of prophylactic vaccine against HEV infection. This review tries to summarize the current knowledge about the efficacy, safety and new strategies to develop HEV vaccines. My major point for the manuscript is within the structure. Is confusing to follow overall for a non-expert reader on the field. I recommend to restructured given more details about clinical trials (I miss safety and long-term efficacy of Hecolin, for example), avidity and comparison between natural and vaccine-induce antibodies (there are evidences in this sense)… In addition I have more comments described below according to specific section.
Introduction
Line 40: “chronic hepatitis among immunocompromised individuals and solid organ transplant recipients”. This sounds redundant, due to in organ transplant recipients acute HEV infection can became in to chronic due to immunosuppression (linked to immunosuppressive drugs use). In fact, the first therapy approach to solve chronic infection in this population is the reduction of immunosuppression.
Line 43: “leading annually to 3.4 million symptomatic cases, 70,000 deaths and 3,000 stillbirths [4]”. Maybe the most appropriate reference for this state is the 2017 WHO Global hepatitis report (available at https://www.who.int/hepatitis/publications/global-hepatitis-report2017/en/ ).
Line 47: Please consider adding references
Line 49: Genotype 8 has been recently described in Bactrian camels (Woo PC et al Emerg Infect Dis 2016; 22:2219–2221).
Line 55-56: “HEV can present as either a non-enveloped virus, existing in faeces and urine, or as a quasi-enveloped virus, existing in serum and cell culture supernatants”. Effectively, the virus can be found enveloped or not. Nevertheless, in sense to be more precise, the virus is non-enveloped in faeces, quasi-enveloped in intestinal cells, and enveloped in blood. Also include references in this sentence.
The design of Hepatitis E vaccines
Lines 97-98: ORF2s and ORF2c differences are critical to well understand the majority of the clinical trials design. I strongly recommend expanding the explanation of ORF2s. In this sense, Figure 1 is not informative. I consider more appropriate to redesign this Figure focusing in structure (Figure 1B could be more adequate in a table including additional information). This would be crucial for potential not expert readers.
Line 141-142: Seems incomplete.
Lines 158-160: More information about clinical trials in general should be included. For example, “Desirable safety and efficacy in a phase II clinical trial study were demonstrated”, what desirable safety and efficacy mean? This expression should be change to more quantifiable data. Additionally, why this project did not progress? This point should be discussed including authors opinion.
The serological evaluation
Table 1 is not informative at all. What authors wants to show with this Table?
Line 349: Burma strain is genotype 1a
Table 2 should include the effect of the neutralization, not only the description of the study.
Table 3: Please include SAR-55, Mex-14 and US-2 in the legend. Also will be great if Genbank accession numbers could be added.
Conclusions and prospects
Lines 447-450: Miss reference
Author Response
We thank the reviewer for his/her careful review of the manuscript. We do appreciate the nice words on the manuscript and the constructive comments on improving the writing. We have made changes to the manuscript based on these comments (with track-change mode), which have resulted in a much improved manuscript. Line to line responses are listed below.
Comments and Suggestions for Authors
Point: Li et al present a review about current knowledge and perspectives of prophylactic vaccine against HEV infection. This review tries to summarize the current knowledge about the efficacy, safety and new strategies to develop HEV vaccines. My major point for the manuscript is within the structure. Is confusing to follow overall for a non-expert reader on the field. I recommend to restructured given more details about clinical trials (I miss safety and long-term efficacy of Hecolin, for example), avidity and comparison between natural and vaccine-induce antibodies (there are evidences in this sense)… In addition I have more comments described below according to specific section.
Response: We thank the reviewer for his/her comments. Clinical results have been summarized in numerous articles. The scope of this review is on the clinical trials (safety, efficacy, etc.), rather than on the analysis of the functional epitopes on the antigen. One aspect related to the clinical trials is the assay on the binding titers or neutralizing titer of the sera derived from clinical trials. However, we do agree that more details about clinical trials will make the description of hepatitis E vaccines more comprehensive. We have added more details about the safety of hepatitis E vaccine in the revised manuscript in Line 172-173. Additionally, the long-term efficacy of Hecolin was shown in Line 178-180. The antibodies induced after HEV infection may not protective. Thus, the antibodies induced by virus infection were not be discussed in this manuscript. That would be a deviation from the main topic of this ‘vaccine-centric’ review.
Point: Line 40: “chronic hepatitis among immunocompromised individuals and solid organ transplant recipients”. This sounds redundant, due to in organ transplant recipients acute HEV infection can became in to chronic due to immunosuppression (linked to immunosuppressive drugs use). In fact, the first therapy approach to solve chronic infection in this population is the reduction of immunosuppression.
Response: We thank the reviewer for the careful review and constructive suggestion. However, we do not agree that the current phrases are redundant. The expression of “chronic hepatitis among immunocompromised individuals and solid organ transplant recipients” if fine as “immunocompromised individuals” are referring to patients with suppressed immune systems due to aging to a disease state such as AIDS, whereas the “solid organ transplant recipients” have deliberately suppressed immune systems. We would like to keep the original sentence.
Point: Line 43: “leading annually to 3.4 million symptomatic cases, 70,000 deaths and 3,000 stillbirths [4]”. Maybe the most appropriate reference for this state is the 2017 WHO Global hepatitis report (available at https://www.who.int/hepatitis/publications/global-hepatitis-report2017/en/).
Response: We thank the reviewer for his/her recommendation. This report has been cited in Line 45 of the revised manuscript as Reference 4.
Point: Line 47: Please consider adding references
Response: We thank the reviewer for his/her careful review. The reference has been added, per suggestion (Reference 6, 7, 8, 9, 10 in the revise manuscript).
Point: Line 49: Genotype 8 has been recently described in Bactrian camels (Woo PC et al Emerg Infect Dis 2016; 22:2219–2221).
Response: We thank the reviewer for his/her professional comment. We have change this sentence into “To date, eight genotypes of HEV have been isolated, and at least five genotypes (genotypes 1-4 and 7) can cause human infection” (in Line 50 of the revised manuscript and cited as Reference 13).
Point: Line 55-56: “HEV can present as either a non-enveloped virus, existing in faeces and urine, or as a quasi-enveloped virus, existing in serum and cell culture supernatants”. Effectively, the virus can be found enveloped or not. Nevertheless, in sense to be more precise, the virus is non-enveloped in faeces, quasi-enveloped in intestinal cells, and enveloped in blood. Also include references in this sentence.
Response: We thank the reviewer for his/her professional and insightful explanation on the forms of hepatitis E virus. More references (17, 18, 19) were added in the section of 2.1 on this important topic. Per suggestion, we revised the sentence as follows:
“HEV was recently found to be a quasi-enveloped virus. It exists as non-enveloped virions in faeces and urine for transmission, whereas its form could be predominantly enveloped in serum for evading neutralizing antibodies”.
Point: Lines 97-98: ORF2s and ORF2c differences are critical to well understand the majority of the clinical trials design. I strongly recommend expanding the explanation of ORF2s. In this sense, Figure 1 is not informative. I consider more appropriate to redesign this Figure focusing in structure (Figure 1B could be more adequate in a table including additional information). This would be crucial for potential not expert readers.
Response: We thank the reviewer for his/her careful review. We agree that ORF2C and ORF2S play an important role in the design of clinical trials. While do we appreciate the Reviewer’s view on the importance of the topic, this review is focused on all vaccine related issues, such as antigen design, epitope analytics and serological assays. The difference between ORF2C and ORF2S is in the first N-terminal 15 aa residues, which is a region quite distant from the peptides used for antigens in the vaccines (Figure 1A & 1B). Too much explanation of ORF2C and ORF2S would be a deviation from the main topic of this manuscript. However, we would like to add the specific information of ORF2C and ORF2S in Figure 1A. The graphical form (Figure 1B) is a better way to show the clinical stages and specific information of three hepatitis E vaccines that have progressed to the clinical development stage. In addition, being presented in a graphical format, it helps the authors to relate to the corresponding structures of the three vaccine antigens.
Point: Line 141-142: Seems incomplete.
Response: We thank the reviewer for his/her careful review. We have changed this sentences into “A 66-aa extension from the N terminus of E2s and could stabilize E2 (aa 394-606) to form hexamers in solution.”(Line 152-153)
Point: Lines 158-160: More information about clinical trials in general should be included. For example, “Desirable safety and efficacy in a phase II clinical trial study were demonstrated”, what desirable safety and efficacy mean? This expression should be change to more quantifiable data. Additionally, why this project did not progress? This point should be discussed including authors opinion.
Response: We thank the reviewer for his/her careful review. We add quantifiable description in the revised manuscript (Line 172-173). Lacking of commercial value might be the main reason why this project did not progress. This point has been added in Line 174 of the revised manuscript.
Point: Table 1 is not informative at all. What authors wants to show with this Table?
Response: Thanks for the comment. The seroconversion induced by vaccines is a quantitative index to be measured. The binding titre in serum samples in an ELISA is always a straightforward method for serological analysis. Notably, a good correlation was observed between the functional and binding titers in mouse sera (He et al. Functional epitopes on hepatitis E virions and recombinant capsids are highly conformation-dependent. Human Vaccines & Immunotherapeutics.2020. Accept) However, different producers may use different coating antigens in their assays. For example, anti-HEV IgG in serum samples from p239-vaccinated humans and rhesus monkeys was analyzed via E2-coated ELISA, while p495-based vaccine may use p495 as coating antigen. The use of different coating antigens makes the cross-lab and cross-product comparison difficult or even impossible. Table1 summaries the different serological assays of different vaccines. For better illustration, the title of Table1 has been changed into “Table 1. Different antigens used in the ELISA methods for supporting the vaccine clinical trials or preclinical development.” HPV serological assays have similar issues and the Pinto et al (Reference56) from NCI are leading the efforts in coming out the solution to enable the cross-lab serological assay comparison.
Point: Line 349: Burma strain is genotype 1a
Response: We thank the reviewer for his/her careful review. We have added this information in Table 2. Thanks again!
Point: Table 2 should include the effect of the neutralization, not only the description of the study.
Response: We thank the reviewer for his/her careful review. Table 2 focuses mainly on the description of the assay format in the Lab. These are the systems that could be potentially used for serological assays in the future. However, the effect of the neutralization of the clinical samples was not tested using virus neutralization assays.
Point: Table 3: Please include SAR-55, Mex-14 and US-2 in the legend. Also will be great if Genbank accession numbers could be added.
Response: We thank the reviewer for his/her reminder. The Genbank accession numbers have been added in the revised manuscript.
Point: Lines 447-450: Miss reference
Response: We thank the reviewer for his/her careful review. The references (80, 81) has been added in the revised manuscript in Line 476-478.
Round 2
Reviewer 2 Report
Authors have faced all my comments and suggestions. I do not have any additional request.